# Techniques of Frameless Robot-Assisted Deep Brain Stimulation and Accuracy Compared with the Frame-Based Technique

**DOI:** 10.3390/brainsci12070906

**Published:** 2022-07-11

**Authors:** Shanshan Mei, Kaijia Yu, Zhiwei Ren, Yongsheng Hu, Song Guo, Yongjie Li, Jianyu Li

**Affiliations:** 1Department of Neurology, Xuanwu Hospital of Capital Medical University, Beijing 100053, China; sophy33@163.com; 2Beijing Institute of Functional Neurosurgery, Xuanwu Hospital of Capital Medical University, Beijing 100053, China; y125153199@163.com (K.Y.); rzw@xwh.ccmu.edu.cn (Z.R.); hys13611342511@163.com (Y.H.); 13581991953@139.com (S.G.); yongjie5305@163.com (Y.L.); 3Department of Neurosurgery, Affiliated Hospital of Nantong University, Nantong University, Nantong 226001, China

**Keywords:** frameless robot-assisted surgery, frame, deep brain stimulation, accuracy, Parkinson’s disease

## Abstract

Background: Frameless robot-assisted deep brain stimulation (DBS) is an innovative technique for leads implantation. This study aimed to evaluate the accuracy and precision of this technique using the Sinovation SR1 robot. Methods: 35 patients with Parkinson’s disease who accepted conventional frame-based DBS surgery (*n* = 18) and frameless robot-assisted DBS surgery (*n* = 17) by the same group of neurosurgeons were analyzed. The coordinate of the tip of the intended trajectory was recorded as x_i_, y_i_, and z_i_. The actual position of lead implantation was recorded as xa, ya, and za. The vector error was calculated by the formula of √(x_i_ − x_a_)^2^ + (y_i_ − y_a_)^2^ + (z_i_ − z_a_)^2^ to evaluate the accuracy. Results: The vector error was 1.52 ± 0.53 mm (range: 0.20–2.39 mm) in the robot-assisted group and was 1.77 ± 0.67 mm (0.59–2.98 mm) in the frame-based group with no significant difference between two groups (*p* = 0.1301). In 10.7% (*n* = 3) frameless robot-assisted implanted leads, the vector error was greater than 2.00 mm with a maximum offset of 2.39 mm, and in 35.5% (*n* = 11) frame-based implanted leads, the vector error was larger than 2.00 mm with a maximum offset of 2.98 mm. Leads were more posterior than planned trajectories in the robot-assisted group and more medial and posterior in the conventional frame-based group. Conclusions: Awake frameless robot-assisted DBS surgery was comparable to the conventional frame-based technique in the accuracy and precision for leads implantation.

## 1. Introduction

Deep brain stimulation (DBS) surgery is an effective treatment for movement disorders such as Parkinson’s disease (PD), dystonia, and essential tremor [1]. The key of this procedure is the accurate implantation of electrodes into the intended nuclei, such as the subthalamic nucleus (STN), the globus pallidus internus (GPi), and the ventral intermediate nucleus (Vim) to maximize therapeutic benefits and minimize potential side effects. In the clinic, all techniques for DBS surgery developed through the years had to be evaluated for accuracy, which can be defined as the radial error in a 2D scan (*x* and *y* axes) or as the vector error in a 3D scan (*x*, *y*, and *z* axes) [2]. The stereotactic frame-based technique for precise image-guided targeting is the gold standard for electrodes implantation and is most widely utilized in DBS surgery at present; it can reach an accuracy of 1 to 3 mm in average [3,4,5,6,7].

The robot-assisted technique for neurosurgery has been used for almost 30 years and has advantages such as an increased accuracy and the property of efficient and reproducible access to targets [8]. Robot-assisted surgery has been used as an innovative technique for DBS and the leads implantation in stereoelectroencephalography (SEEG) in recent years. Experience in robot-assisted DBS surgery was reported in some centers. The leads deviations calculated as the vector error were reported to be from 0.76 mm to 1.6 mm in DBS [9]. Variable implantation techniques among different centers may be responsible for discrepancies of accuracy, such as (1) the utilization of frame-based or frameless techniques, (2) different intraoperative imaging modalities (O-Arm, CT, and MRI), (3) different robots (ROSA, Neuromate, and SurgiScope), and (4) measurements verifying leads positions [9]. In addition, implantation accuracy can be influenced by expertise and proficiency in DBS surgery, which may result in large interinstitutional variances.

Given the few studies exploring the accuracy of frameless robot-assisted surgery compared with frame-based technique [10], we aimed to evaluate accuracy and precision in DBS using these two methods with the same group of neurosurgeons. In addition, the robot Sinovation SR1 (Sinovation, Beijing, China) used in the study is the first neurosurgery robot that has passed the national innovation review and is largely used for neurosurgery in China (Figure 1) [11].

## 2. Methods

### 2.1. Study Population

We included 35 patients (14 males and 21 females, mean age 61.7 ± 10.2 years) with PD who accepted DBS surgery in our single-center from June 2020 to March 2021. Among these patients, 18 patients (7 males and 11 females, mean age 65.3 ± 7.0 years) accepted conventional frame-based DBS and 17 patients (7 males and 10 females, mean age 57.8 ± 11.8 years) accepted frameless robot-assisted Sinovation SR1 (Sinovation, Beijing, China) DBS surgery by a same group of neurosurgeons. We used a uniform procedure for these two techniques in order to decrease system bias. All patients were deemed appropriate candidates for STN-DBS or GPi-DBS after comprehensive preoperative assessments by professional neurologists. The selection of target—either the GPi or STN—was reached by a standard of care interdisciplinary screening and discussion [12]. The study was approved by the Hospital Ethics Board, and informed consent was provided by all patients prior to surgery.

### 2.2. Conventional Frame-Based DBS

The preoperative examination included 3D T1 weighted images (T1w, voxel size 1 × 1 × 1 mm), 3D T2 weighted images (T2w, 0.67 × 0.67 × 0.67 mm), magnetic resonance venogram (MRV), and quantitative susceptibility mapping (QSM) images (voxel size 0.5 × 0.5 × 1), which were scanned on a 3 Tesla magnet (United Imaging Healthcare, 3.0, Tesla, uMR 770, China) several days before the surgery. Patients with severe tremors were on medication or injected with diazepam to eliminate motion artifacts when scanning. On the day of surgery, the CT scan (voxel size 0.49 × 0.49 × 1, Siemens, Erlangen, Germany) was obtained for patients who were head-mounted in a conventional stereotactic Cosman–Roberts–Wells (CRW) frame. The neurosurgeon imported all of these images into the StealthStation navigation system (Minneapolis, MN, USA) and fused preoperative MRI with the CT scan using the rigid-body co-registration. Targets were visually selected on QSM in the anterior commissure (AC)–posterior commissure (PC)-based coordinate system in the Framelink software (S7, Fusion, Medtronic, Minneapolis, MN, USA) on the StealthStation navigation system, and trajectories were designed through an appropriate gyrus, avoiding the intracranial vessels and the ventricular walls. The patients were operated on in a semi-sitting position and under local anesthesia during the procedure of leads implantations. The left electrode was usually implanted first, and then the right electrode. We manually transferred the stereotactic coordinates to the CRW aiming bow. The leads placement (3387, Medtronic) was simulated and rectified using the phantom base prior to implantation. The distal contact at the junction of substantia nigra pars reticulata and inferior border of STN and the second contact was exactly at the junction of zona incerta and dorsolateral area of STN. After the burr hole craniotomy, the lead was implanted through a rigid guiding tube using a microdriver (Alpha Omega, Nazareth, Israel). We did not perform microelectrode recording (MER) as we used a novel MRI sequence of quantitative susceptibility mapping to perform preoperative planning, which can clearly delineate borders of the STN and GPi. In addition, macrostimulation was intraoperatively performed to test sensory and motor thresholds, and to observe ameliorations of symptoms and adverse events. If the amelioration was unsatisfactory or adverse events occurred at a low amplitude, the lead depth would be adjusted by the microdriver or the lead path would be readjusted. For patients who could not cooperate with macrostimulation, intraoperative O-Arm images were scanned to verify lead placements. The lead was anchored with a lead-anchoring system (Stimlock, Medtronic) after macrostimulation, and steps were repeated on the other side. The rechargeable pulse generator (ACTIVA°RC, Medtronic, Minneapolis, MN, USA) was placed in a subcutaneous pocket under general anesthesia in a subsequent process.

### 2.3. Frameless Robot-Assisted Surgery

Preoperative MRIs were performed in line with the frame-based method and were imported to the Sinoplan 2.0 planning software (Sinovation, Beijing, China), which was installed in the robot system. After fusing these images, targets were selected, and bilateral trajectories were designed (Figure 1). On the day of surgery, five bone screws as fiducial marks were implanted into the skull under local anesthesia and a preoperative 3D CT scan was performed (voxel size 0.49 × 0.49 × 1 mm) to allow frameless registration later. In the operating room, the Sinovation SR1 robot (Sinovation, Beijing, China) was secured with the patient’s head via the frame, which was herein used as a fixed connection device. Then, the 3D CT scan was imported to the robot system and was fused with the preoperative planning using the six-degree-of-freedom transformation. Patients were also operated in a semi-sitting position and under local anesthesia during the procedure of leads implantations. The frameless registration was performed through the mechanical contact of the robot probe with five fiducial markers, and the registration accuracy was allowed to decrease below 0.50 mm (Figure 2). A microdriver (Alpha Omega, Nazareth, Israel) with a guiding tube was installed onto the robot arm. Bilateral entry points were marked in a sterile circumstance using the guiding tube through the command movement of the robot arm. After the scalp incision and burr holes were drilled, the dura was perforated using the bipolar electrocautery. Then, the guiding tube was manually placed in alignment with the planned trajectory at a predefined depth to the target through the robot arm. The lead (3387, Medtronic, Minneapolis, MN, USA) was manually implanted after the guiding tube penetrated the parenchyma. Macrostimulation was performed analogous to the conventional frame-based method when the electrode was implanted. If macrostimulation is satisfactory, the lead was anchored with a lead-anchoring system (Stimlock, Medtronic, Minneapolis, MN, USA) and steps were repeated on the other side. The pulse generator was placed in a subcutaneous pocket the same day.

### 2.4. Implantation Accuracy

The position of electrodes was verified by the postoperative 3D CT scan (voxel size 0.49 × 0.49 × 1 mm) performed on the day after surgery. The postoperative CT scan was fused with preoperative MRI in the planning software to verify leads positions by the same neurosurgeon, and this can help the neurosurgeon to select stimulation contact and adjust stimulation parameters later. The stimulator was turned on one week after the surgery, and the parameters were adjusted three months later. The accuracy of lead implantation was calculated as the vector error using the formula of √(x_i_ − x_a_)^2^ + (y_i_ − y_a_)^2^ + (z_i_ − z_a_)^2^. The tip of the trajectory represented the position of the intended target, and coordinates were recorded as x_i,_ y_i_, and z_i_, which represented medial-lateral, anterior-posterior, and superior-inferior distances relative to the midpoint of AC-PC line, respectively. The center of the distal contact of the lead was regarded as the actual lead position and was recorded as x_a_, y_a_, and z_a_ (Figure 3).

### 2.5. Statistical Analysis

The vector error and deviations, including medial-lateral (|x_i_ − x_a_|), anterior-posterior (|y_i_ − y_a_|), and superior-inferior (|z_i_ − z_a_|) distances, were recorded as means ± standard deviation. Comparisons of the vector error and directional deviations between the frame-based technique and the frameless robot-assisted technique were evaluated using independent-samples *t*-tests. Deviations and vector error between the left and right brain hemispheres in the two methods were also compared with independent-samples *t*-tests. Given leads deviations over 2.00 mm are considered meaningful, and the proportion of vector error surpassing the 2.00 mm versus values within the clinically accepted standard (≤2.00 mm) between two groups was analyzed with the χ^2^ test [13,14]. A paired-samples *t*-test was used to analyze the coordinate difference between the intended position and the center of actual distal contact of the lead (x_i_ vs. x_a_, y_i_ vs. y_a_, and z_i_ vs. z_a_). The statistical analysis was performed using the R studio software (Version 1.1.383, Boston, MA, USA) with a *p* < 0.05 for significance.

## 3. Results

DBS surgery was successfully performed in all 35 patients on bilateral sides (*n* = 70 leads). We manually adjusted the depth (1–2 mm) using the microdriver on six leads in the robot-assisted group (17 patients) and on five leads in the frame-based DBS group (18 patients) because tremor and muscle rigidity were unsatisfactorily improved or side effects such as paraesthesia, myoclonus, and dysarthria occurred at a low voltage during macrostimulation. These leads were not included in the analysis of accuracy and precision, because the decision to adjust leads was made according to clinical response and could not reflect the accuracy of respective implantation methods. Intraoperative complications such as intracranial pneumatosis, intracranial bleeding, and electrode dislocation were not observed. Patients in both groups had a satisfactory amelioration of tremor or rigidity during macrostimulation.

The mean vector error between the intended target and actual distal contact of the lead was 1.52 ± 0.53 mm (range: 0.20–2.39 mm) in the robot-assisted group and 1.77 ± 0.67 mm (0.59–2.98 mm) in the frame-based group. The mean directional deviations between the planned and final lead positions are summarized in Table 1. No significant difference was observed in the vector error (*p* = 0.1301) and deviations of coordinates (x, y, and z) between the robot-assisted group and the frame-based group. However, both the mean values and standard deviations of the vector error and deviations of coordinates (x, y, and z) in the frameless robot-assisted group were smaller than the values calculated in the frame-based group (Table 1). Techniques used in DBS surgery were significantly associated with the proportion of vector error > 2.00 mm and ≤2.00 mm (*p* = 0.0255, χ^2^ test). In 10.7% (*n* = 3) frameless robot-assisted implanted leads, the vector error was larger than 2.00 mm with a maximum offset of 2.39 mm, and in 35.5% (*n* = 11) frame-based implanted leads, vector error was larger than 2.00 mm with a maximum offset of 2.98 mm (Figure 4).

In addition, a statistical significance was observed for the mean value of y_i_ vs. y_a_ on bilateral sides in the robot-assisted group, which indicated that the final position of leads was more posterior than expected. In conventional frame-based group, there was a statistical significance for the mean value of x_i_ vs. x_a_ and y_i_ vs. y_a_ on the bilateral sides (Figure 5). This indicated that the actual position of leads was more posterior and medial than the planned trajectory on the bilateral sides in the frame-based group (Figure 6). In both the robot-assisted group and the frame-based group, the vector error and deviations of coordinates (x, y, and z) between the left and right sides were not significantly different for the actual and planned leads positions (Table 2).

## 4. Discussions

Most DBS surgeries are completed on the basis of stereotactic frame at present, which remains the mainstream operation in the world, and this is closely related to its application history, abundant evidence, and advantages. Robot-assisted surgery is an innovative technique for leads implantation in DBS procedure. Data on the accuracy of robot-assisted DBS are limited, and there are discrepancies in certain techniques, which can lead to interinstitutional variances. In this study, we report for the first time awake frameless robot-assisted DBS surgery using the Sinovation SR1 robot (Sinovation, Beijing, China), which is composed of a robot arm, computer control system, and display system, analogous to the ROSA and Neuromate.

The accuracy is defined as the proximity degree of a measured value to the expected value and is indicated as the mean error. Precision is defined as the degree of coincidence of the measured values when this value is repeatedly measured in the same condition and is usually expressed as the standard deviation of the error [15]. In this study, the vector error of the robot-assisted DBS was 1.52 ± 0.53 mm, which met clinical standards of the accuracy. Some centers simply regarded 3 mm as the standard for reimplanting leads, and most studies considered that an accuracy of below 2 mm was ideal for leads placement [16,17]. Holl et al. reported that a deviation of the lead from the intended nuclei beyond 2 mm may induce suboptimal clinical efficacy [18]. Our results are comparable to those obtained using other robot techniques. De Benedictis et al. reported that the accuracy of leads placement using ROSA by the vector error was 1.60 mm for DBS [19]. Varma et al. evaluated the precision of frameless Neuromate-assisted DBS surgeries in 49 patients and reported a mean accuracy of 1.7 mm [20]. Though our results did not reach a higher accuracy, such as a submillimeter grade, their accuracy may further increase along with upgrades of the robot system because Sinovation SR1 is the first-generation robot of Sinovation.

Our results confirm that the accuracy of robot-assisted DBS is equal to results of the conventional frame-based technique with no significant difference in the vector error between the robot-assisted group and the frame-based group. The smaller standard deviation of the vector error in the robot-assisted group compared to the frame-based group indicated a higher precision of robot-assisted technique in electrodes implantation. This can also be demonstrated with the results of a smaller proportion of vector error (10.7%) surpassing the 2 mm threshold in the robot group. The χ^2^ test proved that the techniques used in DBS surgery were significantly associated with proportions of vector error > 2.00 mm and ≤2.00-mm. In addition, our results using the robot-assisted technique were also comparable to results in other centers using the conventional frame-based technique. In 13 patients with movement disorders who accepted Cosman–Roberts–Wells (CRW) frame-based DBS surgeries, a mean of 1.53 ± 0.16 mm vector error was reported for intraoperative CT to verify final lead positions [21], which was similar to the accuracy in our robot-assisted group. Martin Jakobs et al. also reported an accuracy of 2.1 ± 0.6 mm in frame-based DBS surgeries in a large cohort of patients by intraoperative MRI scans, which was higher than our results [22]. Though we have extensive experience with the conventional frame-based system, we did not obtain a higher accuracy of leads placement in the frame-based group than for the robot-assisted group. Our own experience using this innovative frameless robot-assisted technique for DBS was considerable and convincing. Our accuracy and precision may further improve with the upgrade of the robot system and neurosurgeons’ familiarity with this technique. In our experience, the innovative frameless robot-assisted leads implantation offers many advantages: more comfort for patients with no setting of a heavy frame, a shorter operation time, a reliable reproducibility of the position for the arm on a given trajectory, allowing easier adjustments for a trajectory, and avoiding the need to manually set up coordinates. In addition, the robotic technique also has limitations, including a high cost, slow acceptance and popularization, the requirement of many team members, and a long learning curve. However, the economic burden related to surgeons and patients may decrease with the reduction in operation time and simplification of the operation procedure. Meanwhile, for the frame-based technique, lengthy procedures and longer operation times are challenging for the patient and may create additional costs.

In this study, similar results for deviations in medial–lateral, anterior–posterior, and superior–inferior directions were discovered in the two groups. In addition, the results suggest that final leads positions were more medial and posterior than expected on the bilateral sides, which was consistent with the previous study on frame-based stereotactic surgery [23]. In the robot-assisted group, actual leads positions were more posterior than planned positions (Figure 6), and this was consistent with Alice Goia’s result, which also indicated a more posterior position of the final leads [15]. The leakage of cerebrospinal fluid may explain the tendency of posterior deviations when using these two techniques because brain parenchyma and leads tend to shift in the posterior direction due to the supine position when patients undergo the postoperative CT scan [23]. The medial deviations observed in the frame-based group may be due to the gravity of the device for guiding implanting such as the microdriver, and the lateral tilt angle of the planned trajectory can make the tip of the cannula more medial due to the gravity of the guiding device.

However, frameless systems are conceptually closer to robot-assisted procedures, and perhaps a comparison with the latter would have been highly interesting. To date, the stereotactic frame for image-guided targeting remains the gold standard for functional stereotactic operations, and most DBS surgeries were still completed using the stereotactic frame, especially in China. Given that we compared the frameless robot-assisted technique with the conventional frame-based procedure in this study, factors that may influence the accuracy of these two techniques should be analyzed. The lack of quality control mechanisms could be one of the factors that influences the accuracy of frame-based DBS. However, a phantom base can help surgeons manually rectify small mechanical errors prior to leads implantations. The deviations deriving from mechanical errors are unascertainable and are usually rectified depending on the surgeon’s judgment through the naked eye. Because the aiming bow is usually adjusted in most leads’ implantations due to the puncture angle in the frame-based group, accuracy and reliability can be impaired from the surgeon’s subjective decisions (human errors). In addition, procedures of attaching, locking, and wedging mechanical devices, for which surgeons may be neglectful, lack appropriate quality control. Furthermore, errors can arise from manually setting up coordinates, especially when leads are re-implanted and coordinates need to be repeatedly adjusted. Conversely, the robot arm can be commanded to efficiently move between entry points when entry points and targets are selected and the fiducial registration is finished, and this can decrease mechanical errors and human errors. This capability is especially advantageous for adjusting the position of electrodes. It is worth noting that the robotic system in this study is only as a positioning tool to then carry out a manual implantation of the instruments and electrodes. Human errors also existed in the robotic system when we manually implanted the instruments and electrodes. In addition, some mechanical errors exist in the robotic system; its inherent accuracy and precision can be improved by calibration algorithms and reducing human errors.

In addition, the wear and deformation of metallic devices for lead implantation can influence accuracy and reliability over time. The maintenance of a conventional frame-based system is more complicated than the robot system. In the frame-based system, the head frame, screws, phantom base, microdriver, and aiming bow are required to be repeatedly autoclaved after each surgery. However, only a few parts, such as the screws, microdriver, and instrument holders need to be autoclaved in the robot system because the robot itself has no direct contact with the patient during surgery. The long heat exposure and unintended bump on the mechanical devices may create deformation, wear, and dullness, which could be the cause for a lower precision in the frame-based group in this study. However, a more considerate and reliable maintenance of the robot system in the long term may maintain a higher degree of accuracy and precision for leads implantations than the frame-based system.

Some limitations need to be noted in this study. First, we calculated the vector error through comparing the position of the distal contact of final electrodes with the planned position of the tip of electrodes to study accuracy and precision. Medtronic 3387 electrode has a dead space at its tip that can slightly influence its accuracy. However, we calculated the vector error in a same method in the robot-assisted group and the frame-based group, which can counteract this influence when we compare results in two groups. In addition, some surgeons considered that the radial error was a more appropriate measurement to study the accuracy compared to the vector error because the depth of leads may be adjusted depending on the results of macrostimulation. Leads can be inadvertently advanced or retracted when they are deployed and secured, and this can influence the accuracy that is not related to stereotactic methods [2,9]. Second, this study mainly focused on the accuracy of lead placement under different systems and lacks the observation and statistics of clinical data. In our future research, we will further study the stimulation parameters and clinical outcomes in these two groups of patients using different systems for leads implantations, so as to further confirm the advantages of frameless robot-assisted DBS surgery.

## 5. Conclusions

In this study, the awake frameless robot-assisted DBS surgery was comparable to the conventional frame-based technique in the accuracy and precision of leads implantations. The robot-assisted DBS can be an alternative to purely mechanical guidance systems.

## Figures and Tables

**Figure 1 brainsci-12-00906-f001:**
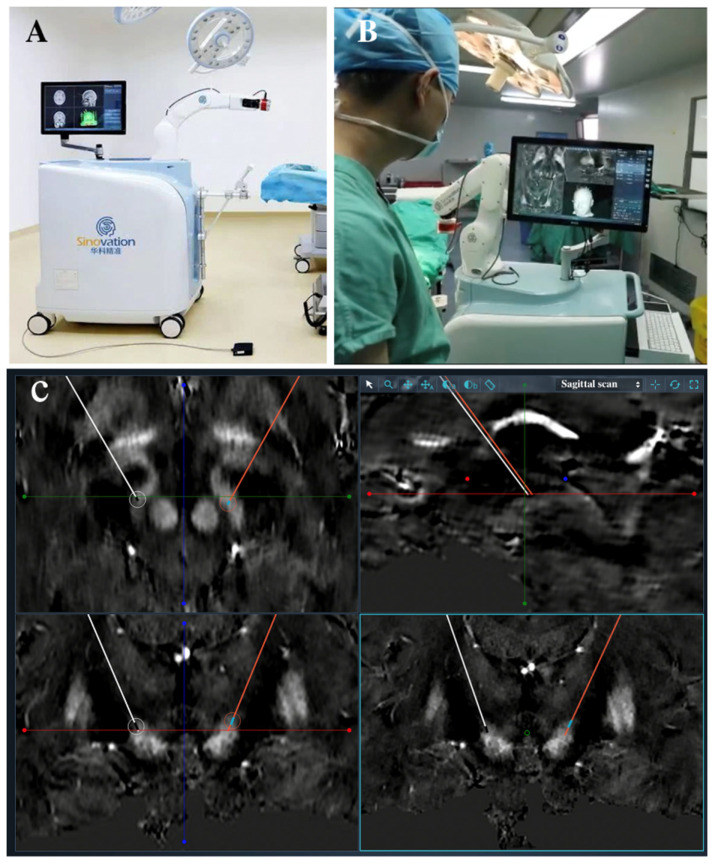
(**A**,**B**) The Sinovation SR1 robot is composed of a robot arm, computer control system, and display system. (**C**) Bilateral trajectories were designed to pass through the subthalamic nucleus (STN) on the Sinoplan 2.0 planning software (Sinovation).

**Figure 2 brainsci-12-00906-f002:**
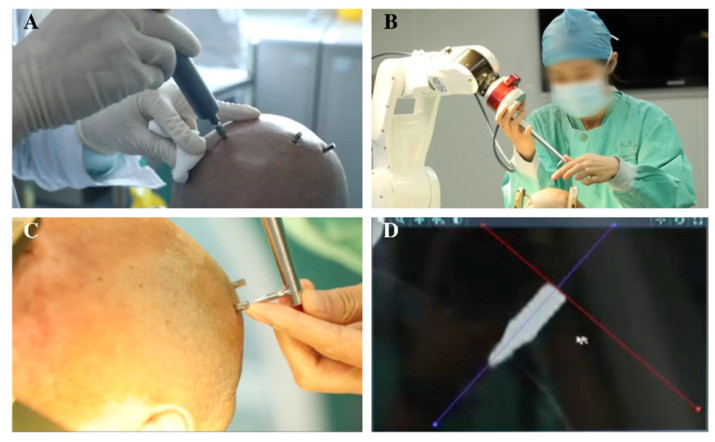
Implanting bone screws and frameless registration. (**A**) Bone screws were implanted under local anesthesia. (**B**–**D**) Frameless registration by the mechanical contact of the tip of the robot probe with bone screw markers.

**Figure 3 brainsci-12-00906-f003:**
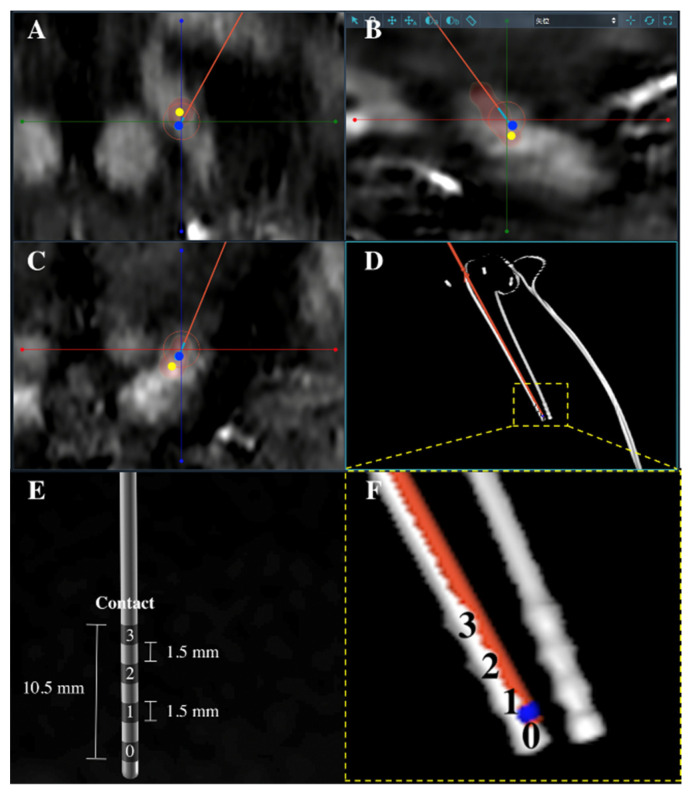
(**A**–**D**) The fusion image of postoperative CT scan with preoperative quantitative susceptibility mapping to display final electrodes positions. Red artifacts (high-density contacts on CT scans were manually set to red) were final positions of contacts of leads. The center of the distal contact (yellow dot) was recorded as x_a_, y_a_, and z_a_ to represent the actual position of leads. The tip of the planned trajectory (blue dot) was recorded as x_i_, y_i_, and z_i_ to represent the intended position of leads. The vector error was calculated by the formula of √(x_i_ − x_a_)^2^ + (y_i_ − y_a_)^2^ + (z_i_ − z_a_)^2^. (**E**) The model of a Medtronic 3387 quadripolar lead has four contacts (the distal contact: contact 0) (**F**) The enlarged view of the planned trajectory (red) and the actual lead implantation (white). The actual distal contact is contact 0.

**Figure 4 brainsci-12-00906-f004:**
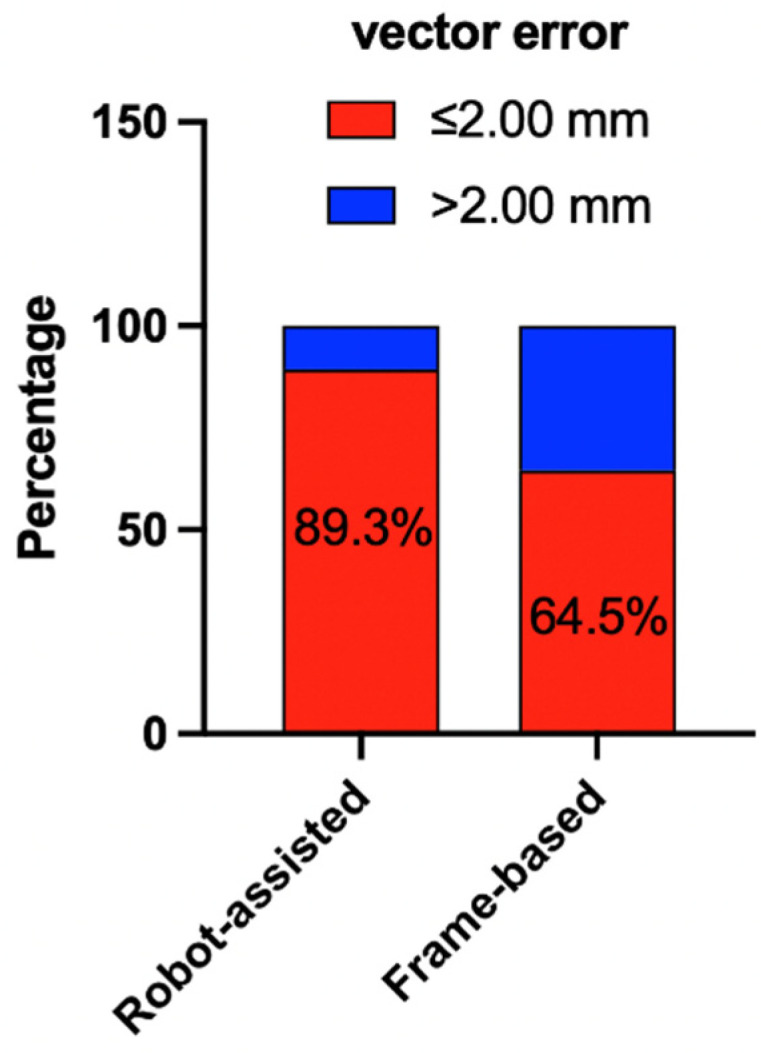
The proportion of the vector error ≤ 2.00 mm and values > 2.00 mm in two groups.

**Figure 5 brainsci-12-00906-f005:**
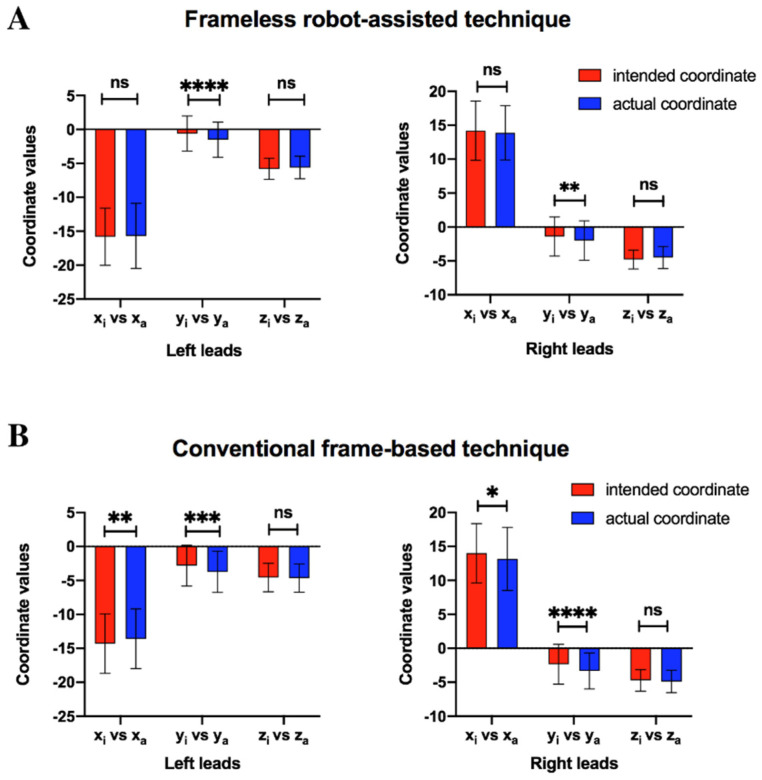
(**A**) A significant difference was observed in the mean value of y_i_ vs. y_a_ in the robot-assisted group. (**B**) A significant difference was observed in the mean value of x_i_ vs. x_a_ and y_i_ vs. y_a_ in the conventional frame-based group. * represents *p* < 0.05, ** represents *p* < 0.01, *** represents *p* < 0.001, **** represents *p* < 0.0001, “ns” represents no significant difference.

**Figure 6 brainsci-12-00906-f006:**
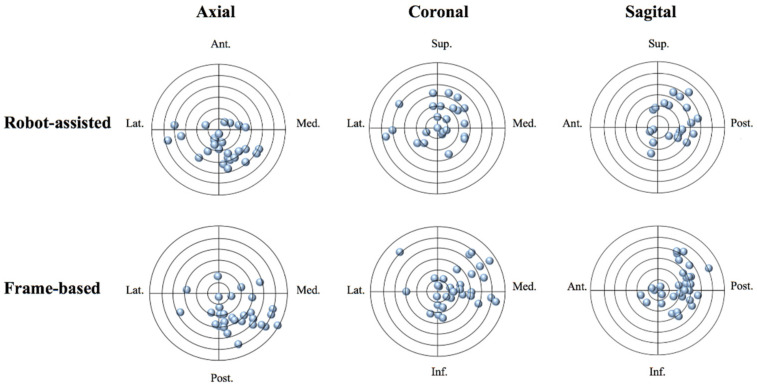
Leads deviations from the planned position in the robot-assisted group and the frame-based group. The planned position of the trajectory is in the center of circles and blue dots are positions of the actual distal contacts of leads implantations. The distance between each one of the circles is 0.5 mm. The final position of leads was more posterior than the planned position using the robot-assisted technique. The final position of leads was more posterior and medial than the planned position using the frame-based technique.

**Table 1 brainsci-12-00906-t001:** Contact deviations and vector error of final leads from the planned position (mm) in two groups.

	∆x_m_	∆y_m_	∆z_m_	Vector Error
Robot-assisted	0.79 ± 0.65	0.80 ± 0.49	0.64 ± 0.48	1.52 ± 0.53
Frame-based	0.99 ± 0.79	1.01 ± 0.51	0.64 ± 0.54	1.77 ± 0.67
*p* value	ns	ns	ns	ns

“ns” in the table represents no significant difference.

**Table 2 brainsci-12-00906-t002:** Comparisons of deviations and vector error on the left and the right side (mm) in two groups.

	Robot-Assisted		*p* Value	Frame-Based		*p* Value
	Left Side	Right Side		Left Side	Right Side	
∆x_m_	0.65 ± 0.78	0.91 ± 0.50	ns	0.90 ± 0.76	1.11 ± 0.85	ns
∆y_m_	0.91 ± 0.49	0.71 ± 0.50	ns	0.97 ± 0.52	1.05 ± 0.52	ns
∆z_m_	0.55 ± 0.45	0.71 ± 0.51	ns	0.61 ± 0.54	0.68 ± 0.56	ns
Vector error	1.53 ± 0.61	1.52 ± 0.48	ns	1.67 ± 0.66	1.88 ± 0.69	ns

“ns” in the table represents no significant difference.

## Data Availability

The research data are available upon request.

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
