# Peer review of "Techniques of Frameless Robot-Assisted Deep Brain Stimulation and Accuracy Compared with the Frame-Based Technique"

_brainsci, 2022, doi:10.3390/brainsci12070906_

Round 1

Reviewer 1 Report

I appreciated the authors' attempt to accurately compare electrode placement for DBS with the two different systems described. I would like to point out that I am an advocate of robotic assisted stereotaxis. There are certainly some critical issues in the paper, however, underlined by the authors themselves, and some insufficient information which, in my opinion, should be elicited both in the methods and analyzed in discussion.

1 - the authors do not fully describe the criteria in the choice of target (STN vs GPi) within the two groups of patients, merely emphasizing that this choice was made by expert neurologists.

2- the authors use 3387 electrodes for all targets. Historically these electrodes were produced for DBS in GPi while Medtronic 3389 has generally been used for the STN. The difference between the two electrodes for spatial dimensions of the contacts is certainly relevant for evaluation purposes. Can the authors clarify the reason for their choice.

3 - often the authors, both in the results and in discussion, underline relationships between DBS and SEEG. In fact, from the stereotaxic point of view, there are differences between the two implantation procedures, especially spatial, which may have a relationship with the accuracy of the positioning - In the DBS (with specific exceptions) coordinating access routes and trajectories require the evaluation of all and three the spatial components while in the SEEG two prevail. This difference needs to be assessed in a more specific way

4 - the verification of the accuracy of the precision of the two systems is not just a geometric comparison between what was programmed and what was actually achieved on the spatial plane and we know very well how these differences are present in a stochastic way in any procedure. Above all, once the procedure has been completed, it is important to know the clinical effectiveness of the lead placement. These data were not sufficiently pointed out in the paper. In my opinion, this aspect should be presented and discussed also given that the comparison between the two methods was reserved for human subjects and does not only have experimental connotations.

5 - I am not an absolute defender of the IOMER's for personal history but could the authors clarify why they were not performed?

6 - as regards the robotic system, was it used only as a positioning tool to then carry out a manual implantation of the instruments and electrodes or did the robotic system also carry out the introduction of the instrumentation? There are experimental studies in the literature that see the execution of the "no hands" procedure using robots. I think it would be useful for the quality of the paper to discuss these aspects as well.

7- no comparison is made with the frameless systems in use was it a deliberate implementation ?? Frameless systems are conceptually closer to the robotic assisted procedure and perhaps a comparison with the latter would have been of undoubted interest.

8- no information was given about the generators subsequently used in the two series of patients and what the stimulation parameters were and whether the differences in positioning influenced the setup of the two series of patients in a more or less significant way.

Author Response

Response to Reviewer 1 Comments

Point 1: the authors do not fully describe the criteria in the choice of target (STN vs GPi) within the two groups of patients, merely emphasizing that this choice was made by expert neurologists.

Response 1: Thank you for your comments. The selection of target—either the GPi or STN—was reached by a standard of care interdisciplinary screening and discussion[1]. (Line 73-76)

Point 2: the authors use 3387 electrodes for all targets. Historically these electrodes were produced for DBS in GPi while Medtronic 3389 has generally been used for the STN. The difference between the two electrodes for spatial dimensions of the contacts is certainly relevant for evaluation purposes. Can the authors clarify the reason for their choice?

Response 2: Though most centers selected 3389 electrodes for STN in order to make more contacts in the nucleus, we used 3387 aiming to make the distal contact at the junction of substantia nigra pars reticulata and inferior border of STN to improve gait, and the second contact was exactly at the junction of zona incerta and dorsolateral area of STN to improve tremor considering the contacts space (1.5 mm) of 3387. (Line 96-100)

Point 3: often the authors, both in the results and in discussion, underline relationships between DBS and SEEG. In fact, from the stereotaxic point of view, there are differences between the two implantation procedures, especially spatial, which may have a relationship with the accuracy of the positioning - In the DBS (with specific exceptions) coordinating access routes and trajectories require the evaluation of all and three the spatial components while in the SEEG two prevail. This difference needs to be assessed in a more specific way.

Response 3: Thank you for your comments. We have deleted contents about the accuracy of SEEG implantation introduced in other studies given that differences between the DBS and SEEG implantation procedures.

Point 4: the verification of the accuracy of the precision of the two systems is not just a geometric comparison between what was programmed and what was actually achieved on the spatial plane and we know very well how these differences are present in a stochastic way in any procedure. Above all, once the procedure has been completed, it is important to know the clinical effectiveness of the lead placement. These data were not sufficiently pointed out in the paper. In my opinion, this aspect should be presented and discussed also given that the comparison between the two methods was reserved for human subjects and does not only have experimental connotations.

Response 4: Thank you for your comments. The clinical effectiveness of the lead placement is actually important. In this study, all patients in both groups had a satisfactory amelioration of tremor or rigidity during the macrostimulation and all patients had beneficial effects in symptom improvements with monopolar or bipolar stimulation mode postoperatively. In our center, we generally adjust stimulation parameters more than 2 times in postoperative 6 months and evaluate symptom improvements using scales and questionnaires. However, some patients cannot return to the hospital due to the COVID-19 Outbreak in recent years, and thus, we have not completed the postoperative evaluation for all patients. We have stated the importance of clinical effectiveness of the lead placement in the limitation part in line 352-356.

Point 5: I am not an absolute defender of the IOMER's for personal history but could the authors clarify why they were not performed?

Response 5: We did not perform microelectrode recording (MER) as we used a novel MRI sequence of quantitative susceptibility mapping to perform preoperative planning which can clearly delineate borders of the STN and GPi. In addition, macrostimulation was intraoperatively performed to test sensory and motor thresholds, and to observe ameliorations of symptoms and adverse events. If the amelioration was unsatisfactory or adverse events occurred at a low amplitude, the lead depth would be adjusted by the microdriver or the lead path would be readjusted. For patients who could not cooperate with macrostimulation, intraoperative O-Arm images were scanned to verify lead placements (line 102-110). In our center, we used IOMER's in early stages to help intraoperatively verifying electrodes placements.In recent years, we only used IOMER's to target Vim or ANT.

 Point 6: as regards the robotic system, was it used only as a positioning tool to then carry out a manual implantation of the instruments and electrodes or did the robotic system also carry out the introduction of the instrumentation? There are experimental studies in the literature that see the execution of the "no hands" procedure using robots. I think it would be useful for the quality of the paper to discuss these aspects as well.

Response 6: Thank you for your questions and suggestions. In this study, the robotic system was used only as a positioning tool to then carry out a manual implantation of the instruments and electrodes. We have discussed these aspects in line 321-324 (It is worth noticing that the robotic system in this study is used only as a positioning tool to then carry out a manual implantation of the instruments and electrodes. Human errors also existed in the robotic system when we manually implant the instruments and electrodes).

Point 7: no comparison is made with the frameless systems in use was it a deliberate implementation?? Frameless systems are conceptually closer to the robotic assisted procedure and perhaps a comparison with the latter would have been of undoubted interest.

Response 7: Thank you for your question. In China, most DBS surgeries were performed using the stereotactic frame, we rarely implant electrodes using the frameless navigation system. In addition, given that stereotactic frame for image-guided targeting remained to be the gold standard for functional stereotactic operations, we compared the robotic-assisted technique with the frame-based technique. We have explained it in line 300-305 (Though frameless systems are conceptually closer to the robot-assisted procedure and perhaps a comparison with the latter would have been of undoubted interest. To date, the stereotactic frame for image-guided targeting remained to be the gold standard for functional stereotactic operations and most DBS surgeries were still completed using the stereotactic frame especially in China. Given that, we compared the frameless robot-assisted technique with the conventional frame-based procedure).

Point 8: no information was given about the generators subsequently used in the two series of patients and what the stimulation parameters were and whether the differences in positioning influenced the setup of the two series of patients in a more or less significant way.

Response 8: Thank you for your suggestion. We used the rechargeable pulse generator (ACTIVA。RC,Medtronic) in both groups (line 111-112). We used the usual parameters as the initial stimulation, monopolar or bipolar stimulation mode with pulse width of 90 us, frequency of 130/160 Hz, voltage of 1.5-3.0 V [2][[3]. What you mentioned is one of the limitations of Our study. We mainly focused on accuracy of lead placement under different systems, and lacks the observation and statistics of clinical data. In our future study, we will further study the stimulation parameters and clinical outcome in these two group patients who using the different systems for leads implantations, so as to further confirm the advantages of frameless robot-assisted DBS surgery.

[1] Higuchi MA, Martinez-Ramirez D, Morita H, Topiol D, Bowers D, Ward H, et al. Interdisciplinary Parkinson's Disease Deep Brain Stimulation Screening and the Relationship to Unintended Hospitalizations and Quality of Life. PLoS One. 2016;11:e0153785.

[2] Kumar R. Methods for programming and patient management with deep brain stimulation of the globus pallidus for the treatment of advanced Parkinson's disease and dystonia. Mov Disord. 2002;17 Suppl 3:S198-207.

[3] Dayal V, Limousin P, Foltynie T. Subthalamic Nucleus Deep Brain Stimulation in Parkinson's Disease: The Effect of Varying Stimulation Parameters. J Parkinsons Dis. 2017;7:235-45.

Reviewer 2 Report

Summary: This study systematically compared the vector error, which is a measure of the accuracy of lead implantation in frame-based and frameless robot-assisted deep brain stimulation surgery. Importantly, this is a carefully done procedure as the authors did not observe any intraoperative complications such as intracranial pneumatosis, bleeding and electrode dislocation. The data is very encouraging as they observed a smaller vector error in the frameless robot-assisted group than in the frame-based group. In my opinion, this study is absolutely timely and I have the following minor comments to improve the presentation.

Comments and Suggestions:

§     In figure 4, instead of showing data as separate bar graphs for <2 mm and >2 mm groups, it is more informative to show them as stacked bars (blue bar on top of red bar).

§    As this is relatively a new adaptation in the clinic for DBS, it will be further useful to provide some insights into the monetary cost of the robotic procedure in comparison with the frame-based method in the discussion section. 

Author Response

Point 1:  In figure 4, instead of showing data as separate bar graphs for <2 mm and >2 mm groups, it is more informative to show them as stacked bars (blue bar on top of red bar).

Response 1: Thank you for your comments and suggestions. We have modified the figure 4 as you suggested. (Line 207-208)

Point 2: As this is relatively a new adaptation in the clinic for DBS, it will be further useful to provide some insights into the monetary cost of the robotic procedure in comparison with the frame-based method in the discussion section.

Response 2:  Thank you for your suggestions. We have added some insights into the monetary cost of the robotic procedure in line 282-288 (In addition, robotic technique also has limitations including its high cost, slow acceptance and popularization, requiring the cooperation of many team members, and a long learning curve. However, the economic burden related to surgeons and patients may decrease with the reduction of operation time and simplification of the operation procedure. While of the frame-based technique, the longer operation time is challenging for the patient and may create additional costs).

Reviewer 3 Report

The authors presented a very well documented and performed research for comparing a frameless robot-assisted group of Parkinson-affected patients against a similar cohort intervened under the traditional frame-based technique. I’m having, however, some concerns about the statistical techniques used and the conclusions achieved.

Please, clarify the following:

·         Section 2.5. Line 159: The authors state to have recorded the deviations using means and corresponding standard deviations. However, the next claim is to have used non-parametric tests which deal with medians. Please clarify.

·         Furthermore (line 163), the authors implemented a non-parametric test aimed to compute the differences between two groups only. Thus, I don’t see the reason as to why the Dunn’s post-hoc test is needed. Similar argument can be applied for the Friedman test as this the non-parametric alternative for the one-way ANOVA with repeated measures.

·         Line 165: the use of Kruskal-Wallis test involves ranked data. I can’s see the idea of ranked data within the variables used. Please clarify.

·         Line 188: Although the authors state to have found differences between the robot-assisted and frame-based groups, no statistical tool seems to have been used to back this up.

·         Line 191: Authors do mention to have used the chi-square test. This is a test to assess relationships or associations, not differences. Please explain.

Author Response

Point 1:    Section 2.5. Line 159: The authors state to have recorded the deviations using means and corresponding standard deviations. However, the next claim is to have used non-parametric tests which deal with medians. Please clarify.

 Furthermore (line 163), the authors implemented a non-parametric test aimed to compute the differences between two groups only. Thus, I don’t see the reason as to why the Dunn’s post-hoc test is needed. Similar argument can be applied for the Friedman test as this the non-parametric alternative for the one-way ANOVA with repeated measures.

 Line 165: the use of Kruskal-Wallis test involves ranked data. I can’t see the idea of ranked data within the variables used. Please clarify.

Response 1: Thanks for pointing out the inappropriate statistical method used in the study. We have reviewed data and made statistics again using the parametric test. We compared the vector error and directional deviations between the frame-based technique and the frameless robot-assisted technique using independent-samples t-tests. Deviations and vector error between the left and right brain hemispheres in two methods were also compared with independent-samples t-tests. Dunn’s post-hoc test and the Friedman test were not needed. We have rewritten the “Statistical analysis” part in line 167-179 (The vector error and deviations including medial-lateral (|xi − xa|), anterior-posterior (|yi − ya|), and superior-inferior (|zi − za|) distances were recorded as means  standard deviation. Comparisons of the vector error and directional deviations between the frame-based technique and the frameless robot-assisted technique were evaluated using independent-samples t-tests. Deviations and vector error between the left and right brain hemispheres in two methods were also compared with independent-samples t-tests. Given leads deviations over 2.00 mm are considered meaningful, the proportion of vector error surpassing the 2.00-mm versus values within the clinically accepted standard (≤2.00 mm) between two groups was analyzed with the χ2 test [12, 13]. A paired-samples t-test was used to analyze the coordinate difference between the intended position and the center of actual distal contact of the lead (xi vs xa, yi vs ya, and zi vs za). The statistical analysis was performed using the R studio software (Version 1.1.383) with a p < 0.05 for significance).

Point 2: Line 188: Although the authors state to have found differences between the robot-assisted and frame-based groups, no statistical tool seems to have been used to back this up.

Response 2: Thanks for your comments. The means of vector error and directional deviations between the robot-assisted and frame-based groups have no statistical difference. As the stereotactic frame for image-guided targeting remained to be the gold standard for DBS surgery. The similar means of vector error in two groups was significative, and this indicated that the frameless robot-assisted DBS surgery was comparable to the conventional frame-based technique in the accuracy. In addition, the standard deviation of the vector in the robot-assisted group was smaller than results in the frame-based group. This suggested a higher precision of the robotic technique compared to the frame-based technique. The χ2 test indicated that techniques used in DBS surgery were significantly associated with the proportion of vector error > 2.00-mm and ≤2.00-mm (p = 0.0255, χ2 test), and this also suggested a higher precision of the robotic technique. We have modified the discussion and the conclusion. We thought the conclusion of “In this study, the awake frameless robot-assisted DBS surgery was comparable to the conventional frame-based technique in the accuracy and precision for leads implantations. The robot-assisted DBS can be an alternative to purely mechanical guidance systems” may be a rigorous conclusion.

Point 3: Line 191: Authors do mention to have used the chi-square test. This is a test to assess relationships or associations, not differences. Please explain

Response 3:  Thanks for your comments. We used the χ2 test to explore whether techniques (robot-assisted or frame-based) used in DBS surgery were associated with the proportion of vector error > 2.00-mm, because the vector error surpassing the 2-mm threshold is significant in DBS. Neurosurgeons considered that a beyond 2-mm deviation of the lead from the intended nuclei may induce suboptimal clinical efficacy in DBS, thus they would reimplanted the electrodes if a beyond 2-mm vector error existed after leads placement. We have modified the description in line 200-201 (Techniques used in DBS surgery were significantly associated with the proportion of vector error > 2.00-mm and ≤2.00-mm (p = 0.0255, χ2 test) and line 264-265 (The χ2 test proved that techniques used in DBS surgery were significantly associated with proportions of vector error > 2.00-mm and ≤2.00-mm).The figure 4 was also rectified in line 207-208.

Round 2

Reviewer 3 Report

I appreciate the authors taking the time to re-assess and improve the analysis of the study.